# Structural basis for activation of a diguanylate cyclase required for bacterial predation in *Bdellovibrio*

Richard W. Meek[1], Ian T. Cadby [1], Patrick J. Moynihan [1] & Andrew L. Lovering [1]

The bacterial second messenger cyclic-di-GMP is a widespread, prominent effector of life-style change. An example of this occurs in the predatory bacterium *Bdellovibrio bacteriovorus*, which cycles between free-living and intraperiplasmic phases after entering (and killing) another bacterium. The initiation of prey invasion is governed by DgcB (GGDEF enzyme) that produces cyclic-di-GMP in response to an unknown stimulus. Here, we report the structure of DgcB, and demonstrate that the GGDEF and sensory forkhead-associated (FHA) domains form an asymmetric dimer. Our structures indicate that the FHA domain is a consensus phosphopeptide sensor, and that the ligand for activation is surprisingly derived from the N-terminal region of DgcB itself. We confirm this hypothesis by determining the structure of a FHA:phosphopeptide complex, from which we design a constitutively-active mutant (confirmed via enzyme assays). Our results provide an understanding of the stimulus driving DgcB-mediated prey invasion and detail a unique mechanism of GGDEF enzyme regulation.

[1] Institute for Microbiology and Infection, School of Biosciences, University of Birmingham, Birmingham B15 2TT, UK. Correspondence and requests for materials should be addressed to A.L.L. (email: a.lovering@bham.ac.uk)

The switch between distinct lifestyle states in bacteria (e.g. motile to sessile, biofilm to planktonic) is often co-ordinated by the ubiquitous second messenger cyclic-di-GMP (c-di-GMP)[1]. Therein, sensory subdomains (commonly receptors for small molecules, or domains gated by protein:protein interaction) control the activity of c-di-GMP production and degradation by appending these to GGDEF and EAL/HD-GYP enzymatic domains, respectively—all three enzymes named after active site motifs. Free c-di-GMP is sensed by a variety of recognition modules, e.g., PilZ domains[2], riboswitches[3], non-catalytic GGDEF/EAL variants[4], and coupled to downstream outputs that effect the lifestyle switching. Different lifestyles require varying pathway complexity—one such bacterium with a high c-di-GMP "intelligence" is the bacterial predator *Bdellovibrio bacteriovorus*, encoding 5 GGDEF synthases (one degenerate), 1 EAL and 6 HD-GYP hydrolases (4 degenerate), and >18 PilZ receptors in the model strain HD100 (ref. [5]). This network complexity may run to even greater depths, with an additional 65 putative novel receptors identified using c-di-GMP resin affinity methods[6]. This need for multiple, diverse signaling pathways is presumably related to the diverse environments encountered during the *Bdellovibrio* lifecycle.

*B. bacteriovorus* is an obligate bacterial predator of other Gram-negative bacteria, exploiting adaptations that allow con-sumption from inside the prey cell periplasm[7]. Access to this "private dining niche" is afforded by a staged invasion process requiring (i) a free-living phase where the predator swims and/or glides to locate prey; (ii) an initially reversible attachment event that proceeds to a tight junction between both cells; (iii) whole-cell outer-membrane invasion, in which the predator pulls itself into the prey periplasm using a type IV pilus; and (iv) resealing of the invasion pore. The invaded cell is stabilized and modified to become what is referred to as the bdelloplast, prior to metabolism and assimilation of prey macromolecules by *Bdellovibrio*. The filamentous predator then grows and septates, releasing progeny via lysis of the host cell. Landmark studies revealed discrete phenotypes for *Bdellovibrio* GGDEF gene knockouts, and that each GGDEF could not compensate for loss of another[8]. The four GGDEFs shown to be active (linking two molecules of GTP to give the c-di-GMP product) were given the prefix Dgc (Digua-nylate cyclases) A-D, and the remaining degenerate variant GGDEF was found to be a receptor (CdgA). The varied pheno-types tallied with lifecycle events, with loss of DgcA preventing exit from the exhausted prey shell, and loss of CdgA resulting in slower invasion than wild-type cells. Intriguingly, loss of DgcB and DgcC gave rise to opposing effects—the ΔDgcB predator unable to commit to prey invasion (resulting in obligate axenic growth), and the ΔDgcC predator unable to convert to a form able to grow outside prey (host-independent stage, licensed in the lab via mutation). ΔDgcD cells had no gross observable deficit and require further investigation to ascertain any functional importance. Hence, it was hypothesized that lack of natural complementation between Dgc knockouts resulted from either local circuits of c-di-GMP (resulting in a signal not freely diffu-sible) or a requirement for the Dgc proteins to be physically present at a cellular site. An example of such a system has been observed in other bacteria, e.g. *Pseudomonas fluorescens*, where c-di-GMP has been shown to bind to an inhibitory (I-site, RxxD motif) region of the GGDEF fold, turning the resulting protein:nucleotide complex into a unique signaling agent[9].

Control of initiation of the *B. bacteriovorus* predation lifecycle may thus involve the opposing effects of DgcB and DgcC, with invasion gated by prey-contact/encounter-stimulated activation of DgcB. Activated DgcB would then signal to CdgA, amongst other targets. The ability to invade and kill prey (as monitored by plaque formation on prey lawns) was demonstrated to be dependent on active DgcB, given that the knockout could not be complemented by a GGAAF non-catalytic variant[8]. The *dgcB* gene, *bd0742*, has orthologues in other *Bdellovibrio* strains and also distantly related myxococcal predators (that kill prey by an entirely different "Wolfpack" mechanism)[10]. The DgcB protein appends the C-terminal GGDEF enzymatic domain with an N-terminal forkhead-associated (FHA) domain, known to be used (in other proteins) for interactions with phosphopeptide ligands[8]. The novel domain architecture of DgcB is interesting—were the activity of DgcB to be dependent on classical FHA function/principles, invasion could potentially involve a sensory kinase that would respond to prey encounter by phosphorylating a protein ligand which would then bind DgcB at the FHA domain to regulate GGDEF activity and thus c-di-GMP production. We reasoned that a structure of DgcB would inform on whether the predicted N-terminal FHA domain was a bone-fide signaling domain with a consensus binding cleft, and whether its juxta-position to the GGDEF domain was likely to render it inhibitory or stimulatory.

To determine the precise mechanism behind DgcB activation, with the aim of understanding the events that license switching of *Bdellovibrio* into an invasion-competent diguanylate cyclase-active state, we determined the structure of DgcB. Our structure reveals a stimulatory role for the N-terminal tail of DgcB and we show that the DgcB FHA domain recognizes this tail in its phosphorylated state. In addition, we confirm this mechanism of activation by the use of disulfide mutagenesis to lock DgcB in an obligate-activated state.

## Results

**DgcB structure determination**. An initial full-length construct yielded a 1.79 Å crystal structure (statistics for all structures are provided in Table 1), composed of two copies in the asymmetric unit. This structure was solved via molecular replacement using isolated domains from GGDEF and FHA proteins, allowing the majority of the polypeptide chain to be resolved (aa 32–134 and 144–310). DgcB comprised a ~30aa tail, 100aa FHA, 20aa linker and 160aa GGDEF domain. Distance constraints guided the assignment of FHA domains to their respective GGDEF partners, revealing a unique conformation for each chain. This arrange-ment is best described as a pseudosymmetrical GGDEF dimer, appended to two FHA domains that are asymmetrically orien-tated (Fig. 1a); the net effect is to place the FHA domains closer to GGDEF chain B. The dimer is stabilized by two molecules of c-di-GMP (classically intercalating via their planar bases), bound at the I-site of both monomers (Fig. 1a, b). The bound c-di-GMP arises from co-purification, as a result of (basally) active DgcB. The resulting (FHA:GGDEF:c-di-GMP)$_2$ DgcB complex has approximate dimensions of $70 \times 70 \times 40$ Å. The overall DgcB architecture and topology is unique among determined structures but the individual domains are in good agreement with char-acterized domains from other proteins. DALI analysis[11] of structural similarity results in high confidence matches of DgcB with the GGDEF domain of PleD (2V0N[12], Z-score 26.3, 35% identity, RMSD 1.3 Å for 162aa alignment) and the FHA domain of GarA (2XT9, Z-score 18.5, 34% identity, RMSD 0.9 Å for 93aa alignment), among many other significant pairwise agreements.

**The DgcB FHA domain has a consensus structure**. DgcB retains the archetypal ~100aa 11-stranded β-sandwich topology of the FHA superfamily (6-stranded antiparallel sheet, 5-stranded mixed sheet), used by varying proteins to interact with pThr containing peptides. FHA domains are usually compact, recognizing a linear phosphopeptide substrate in their binding partner, e.g. the FHA domain of Rad53p forms a complex with a phosphopeptide from

**Table 1 Data collection and refinement statistics**

| | Full length | FHA:tail | FHA: phosphopeptide |
|---|---|---|---|
| Accession code | 6HBZ | 6HC0 | 6HC1 |
| Data collection | | | |
| Space group | P6₅22 | P2₁2₁2₁ | P6₅22 |
| Cell dimensions | | | |
| $a, b, c$ (Å) | 176.8, 176.8, 112.1 | 69.4, 69.4, 129.0 | 68.8, 68.8, 191 |
| $\alpha, \beta, \gamma$ (°) | 90, 90, 120 | 90, 90, 90 | 90, 90, 120 |
| Resolution (Å) | 1.79 (90.45)ᵃ | 1.87 (36.53) | 1.49 (56.84) |
| $R_{sym}$ | 0.120 (>1.0) | 0.079 (>1.0) | 0.129 (>1.0) |
| $R_{pim}$ | 0.028 (0.455) | 0.033 (0.777) | 0.030 (0.792) |
| $I/\sigma I$ | 20.4 (2.6) | 19.1 (1.4) | 16.1 (1.2) |
| CC ½ | 0.998 (0.448) | 1.0 (0.617) | 0.999 (0.545) |
| Completeness (%) | 100 (99.6) | 100 (100) | 100 (100) |
| Redundancy | 37 (36.9) | 12.8 (12.6) | 33.5 (24.2) |
| Refinement | | | |
| Resolution (Å) | 1.79 | 1.87 | 1.49 |
| No. of reflections | 97426 | 52162 | 44616 |
| $R_{work}/R_{free}$ | 17.9/21.6 | 17.7/20.6 | 14.2/17.8 |
| No. of atoms | | | |
| Protein | 4192 | 3149 | 1656 |
| Ligand/ion | 113 | 3 | — |
| Water | 439 | 284 | 192 |
| B-factors | | | |
| Protein | 35.74 | 41.95 | 26.37 |
| Ligand/ion | 32.48 | 37.92 | — |
| Water | 44.37 | 48.01 | 40.29 |
| R.m.s. deviations | | | |
| Bond lengths (Å) | 0.011 | 0.017 | 0.016 |
| Bond angles (°) | 1.59 | 1.79 | 1.70 |

ᵃValues in parentheses are for highest-resolution shell. Each dataset derives from a single crystal

Rad9p, and is used to regulate the cell cycle of yeast[13]. The FHA domain of DgcB is relatively small in size (with no significant insertions into the loops between strands), resulting in a very compact subdomain with "flat" faces. Consistent with assignment to the FHA superfamily, the DgcB sequence contains the consensus residues that comprise the ligand-binding cleft, using S75, R76 and T96 at the apical face of the FHA domain to create a classical phosphopeptide recognition pocket[14]. The floor of this pocket is stabilized by the buried H78 sidechain. Examination of the pocket reveals that the putative phosphate site is occupied by a bound ion in chain B, and that of chain A is partially occluded by residue D56 (Fig. 1c). The D56 interaction between one FHA domain and another sits at the centre of an interface that buries 582.4 Å² surface area. The FHA:FHA interface is composed of both hydrophobic and hydrophilic interactions, packing the phosphopeptide binding surface/loops of chain A against one face of the chain B β-sandwich. The vast majority of contacts to the GGDEF dimer are made by the chain A FHA domain (Fig. 1d); these comprises both polar (D70 and K48 to chain B) and non-polar (I40 and I126 to chain A, Y45 and Y68 to chain B) interactions. A putative N-terminal tail preceding the FHA domain (residues 1–31) is present on our construct but presumed disordered in this structure.

**Features of linker region and enzymatic GGDEF domain.** The interdomain region (residues 135–143 inclusive) is not discernable for either chain in the electron density map, needing to span a distance of 14/19 Å for chains A/B, respectively. The remaining linker polypeptide is in differing conformations between the two monomers: chain A has a small helical region in the linker (146–154, displaying density for the mainchain but weak density for sidechains), and chain B has an extended conformation that starts the first GGDEF helix (α0, 157–172) one turn earlier than chain A. The chain B arrangement is more comparable to other determined GGDEF structures[1,12], wherein the preceding conserved DxLT motif (150–153) involved in enzyme activation is present in a tight turn (in chain A this has a differing structure and is part of the small helix). Other than the linker region, the two GGDEF domains are very similar in structure, with the active site motifs (sequence GGEEF, natural D to E variant) facing in opposing directions (Fig. 1).

The bound c-di-GMP dimer at the DgcB I-site is orientated in line with the two-fold axis of the GGDEF dimer (Fig. 1b), complexed by R218 and D221 of the conserved RxxD motif. Additional contacts to the c-di-GMP are made from residues T242, V246 and R249 on the face of an α-helix. The sidechain of R249 stacks with the guanine base, hydrogen bonding to the O and N7 of the opposing c-di-GMP guanine and the carbonyl of L216. This interaction appears to stabilize a helical turn that precedes the RxxD motif into a $\pi$ (I +5) conformation, and allow the carbonyl of K215 to hydrogen bond to the c-di-GMP 2′-OH. The interactions of DgcB with the c-di-GMP ligand provide the major contact between the two GGDEF domains (Fig. 1b); direct protein:protein contacts are limited to just one weak interaction involving the partially ordered linker region.

**A FHA-only structure suggests a self-stimulatory peptide.** Our "full-length" structure of DgcB did not reveal the conformation of the N-terminal tail (residues 1–31), possibly as a result of being locked in a c-di-GMP-inhibited state. To resolve this issue, we obtained 1.87 Å data on a new crystal form (tail-FHA only, lacking the GGDEF domain) comprised of residues 1–135 of DgcB. In this new form, four FHA domains comprise the asymmetric unit, with each monomer arranged at ~90° to the neighboring domain. Upon imposing crystal symmetry, it is apparent that each monomer contacts a symmetry-related molecule in precisely the same manner as chains A and B of the full-length DgcB structure (mediated through D56, Fig. 2a, also Supplementary Fig. 1). Hence it can be inferred that the FHA:FHA contacts in full-length DgcB are not dictated by constraints imposed by the GGDEF domain.

For one of the FHA domains, difference density was observable at the putative phosphopeptide-binding cleft (next to consensus residues S75, R76 and T96). Our high resolution data allowed us to unambiguously model this density as residues K13-S19 of the previously disordered N-terminal tail of DgcB. It was not possible to assign the tail as originating from a specific chain (chain B, C and D FHA domains are all close enough to span the distance to the bound tail). In agreement with assignment of the FHA domain as a putative phosphothreonine receptor, the interactions between the tail and FHA cleft centre around residue T14, which is conserved in homologues of DgcB (Fig. 2b). The FHA cleft makes three hydrogen bonds to the tail (involving R61, T96 and N97), which binds in a linear conformation across the apical face of the FHA fold (Fig. 2a, c). Prior to this study, any potential stimulus peptide substrate for DgcB was unknown—our observation that the FHA of DgcB could recognize a self peptide led us to investigate sequence features and conservation at this region of the protein. Aligning the N-terminal tail from a representative group of DgcB homologues reveals a pattern of conservation across a seven residue span (−3 to +3 with respect to the central threonine); our structure underlines the importance of the +3 position (V17), which points downward into the cleft and is a

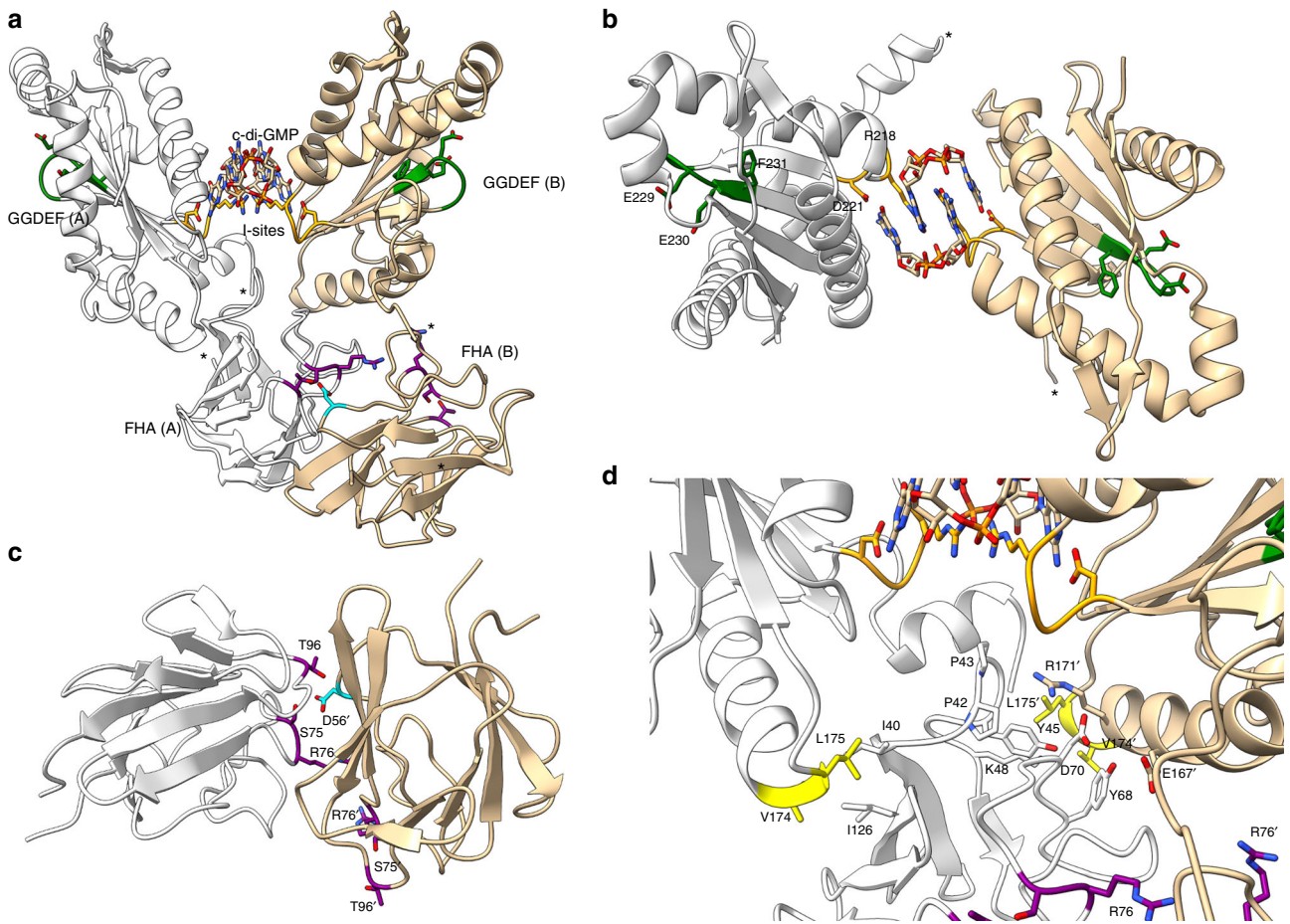

**Fig. 1** Structure of full-length *B. bacteriovorus* DgcB (I site RxxD motif colored orange, active site GGEEF sequence green), with bound c-di-GMP located at interface of the two chains (A white, B tan, residue numbers in chain B given as prime). Residues in the FHA domain binding cleft are colored purple. **a** Juxtaposition of asymmetric FHA domains and GGDEF dimer, end of traceable regions denoted by sphere and asterisk symbol. **b** View 90˚ to that of **a**, demonstrating c-di-GMP intercalation at I-site of GGDEF dimer, holding the two active sites (green) antipodal to one another. **c** View from beneath GGDEF interface, showing FHA:FHA interaction and the projection of D56 (cyan) from chain B into the chain A FHA binding cleft. The two FHA domains are related by a ~90˚ rotation. **d** The chain A FHA domain is sandwiched by both protomers of the GGDEF dimer, particularly the helical turn residues V174 and L175 (colored yellow)

crucial recognition feature of the peptide substrate in other FHA-based sensory systems[14].

The highest-confidence matches (via DALI) of the DgcB FHA are GarA and OdhI, which also utilize a self-peptide recognition mode for signaling purposes (using steric occlusion to regulate interaction with other proteins[15,16]). Comparison of the tail region between DgcB and homologues suggests that similar features are recognized between the two differing systems: hydrophobic residues at the −3, +2 and +3 positions, glutamate (DgcB E12) at −2 and serine (S15) at +1. Upon structural superimposition of DgcB and OdhI via the FHA β-sandwich, it is apparent that the phosphopeptide of the OdhI tail sits in a slightly different position to the unphosphorylated tail of DgcB (Fig. 2d). In total, our structures and the sequence conservation of the DgcB tail suggest that this region is likely to be the true ligand for the FHA domain.

**A phosphopeptide complex validates the DgcB:tail interaction**. We sought to validate the in-cis FHA:tail interaction (a "basal" unphosphorylated state) via obtaining a structure of the (likely more relevant) phosphopeptide complex. By combining a tailless DgcB FHA construct (residues 32–131, V32 replaced with an initiating methionine) and synthetic phosphopeptide

(N[7]SDNLEKpTSIVASDT[17]) in-trans, we were able to obtain complex crystals diffracting to 1.48 Å resolution. The asymmetric unit of this new form contained two FHA domains that were in an offset dimer identical to both the full-length and FHA-only structures, providing a third independent observation of this interaction. The exposed FHA cleft of chain B displays clear density for a bound phosphopeptide, allowing unequivocal modeling of residues 11–21 (Fig. 3a–c). The conformation of the bound tail region does not differ substantially from that of the unphosphorylated complex (rmsd 1.0 Å for all atoms), but sits relatively deeper into the FHA cleft, making more extensive contacts (Fig. 3d). The more deeply situated tail is now able to form mainchain hydrogen bonds with R61, K72 and N97 (via both groups of the amide sidechain). The extended conformation of the tail (~30 Å across) results in the sidechains of E12, K13, I16 and S19 facing away from the FHA domain. The FHA surface creates two depressions which accommodate the two "inwardly projected" residues of the tail (pT14 and +3 V17, Fig. 3b); the flanking L11 and T21 sidechains sandwich opposing sides of the central FHA body. The C-terminal end of the tail packs against the preceding residues, with the sidechain of T21 making a hydrogen bond to the carbonyl of A18. The central pT14 of the peptide tail is recognized by a multitude of interactions, the sidechain methyl group projecting into a pocket formed by

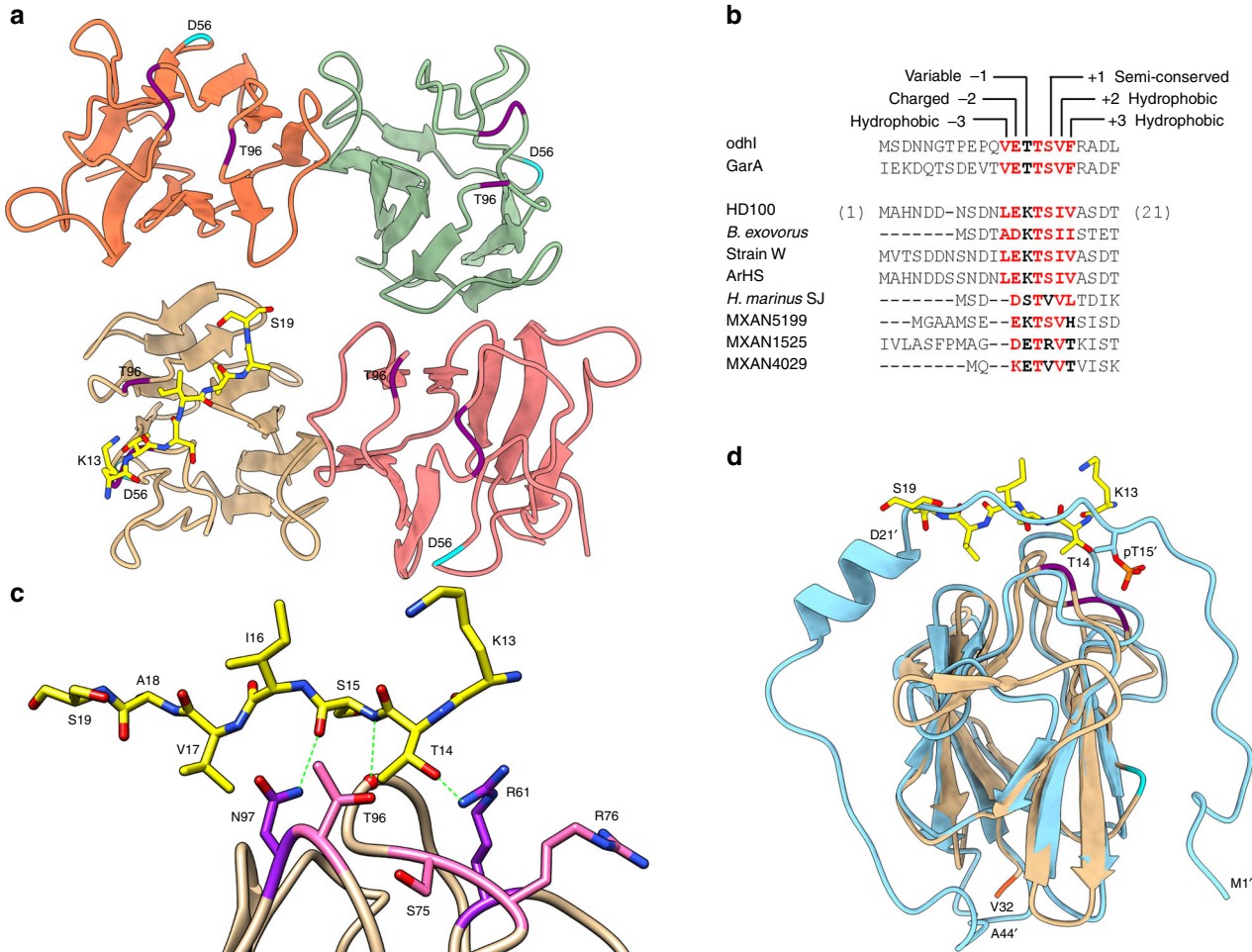

**Fig. 2** Structure and analysis of a DgcB tail-FHA construct lacking the GGDEF domain. **a** The asymmetric unit of this form contains four FHA domains, arranged at ~90˚ to each other, with each replicating the D56-mediated FHA:FHA interactions of the full-length structure (via symmetry mates at edge of tetramer, not shown in figure). One of the four tail regions is ordered (peptide K13-S19, colored yellow) binding across the recognition cleft of a FHA domain, suggestive that the DgcB peptide ligand is derived from self. **b**) Sequence alignment of N-terminal tails of known self-interacting FHA domains (Mycobacterial GarA and OdhI) with DgcB and selected homologues (DgcB equivalents from five *Bdellovibrio* strains and three similar *M. xanthus* FHA: GGDEF proteins). The properties of six positions (−3 to +3) flanking the central Thr of the binding motif are annotated. **c** Interaction between the DgcB tail peptide and FHA domain, hydrogen bonds shown as green dashed lines. Peptide recognition is driven by two groupings of residues that line a cleft of the FHA domain (near grouping colored pink, far grouping purple; cleft is easily discerned in **a** where both groups are colored purple). **d** Structural overlay of DgcB FHA:tail complex (FHA tan, tail yellow) with that of OdhI (blue, PDB code 2KB3, residue numbers denoted by prime). The peptide ligands of both DgcB and OdhI bind in a similarly extended conformation, with DgcB T14 and OdhI pT15′ projecting into a conserved pocket on their respective FHA domain receptors

residues 72–75 and 96–97, which positions the phosphate toward the positively charged R61 and R76 (Fig. 3). The phosphoryl group is tightly co-ordinated by the FHA cleft of DgcB: OG by R61, O1 by S95 and T96, O2 by both the backbone NH and sidechain of R76, and O3 by R76 (and the backbone NH of pT14 also). Comparison of the peptide-bound and unbound FHA domain structures reveals that peptide binding does not induce any obvious gross conformational change or long-range signaling motions, merely alterations in the rotameric states of R61, K72 and R76 (Fig. 3d). These alterations do not seemingly propagate any further change in the molecule distant from the recognition cleft (Supplementary Fig. 1).

**Design of a constitutively active mutant protein**. Our structure of the phosphopeptide tail:FHA complex was suggestive of a model wherein phosphorylation of the DgcB tail could activate the C-terminal GGDEF domain toward producing c-di-GMP

(with the above observations suggesting that activation would arise via domain rearrangement). To test this hypothesis, in the context of not having identified the *Bdellovibrio* kinase that could specifically phosphorylate the tail, we aimed to use our structure to design a mutant with an enhanced propensity to adopt the tail-bound conformation. Initially we attempted a phosphomimetic mutant: activity assays with a T14D-R218A variant (the latter substitution preventing feedback inhibition at the I site) gave only a very small increase in diguanylate cyclase activity c.f. wild-type protein (Fig. 4), in agreement with the general literature view that acidic substitutions are not good analogues of phosphothreonine. We therefore sought to identify residues that would tolerate disulphide linkage between the tail and body of the FHA domain (one cysteine substitution on each, adjacent to one another), giving a covalent "trapping" under oxidizing conditions. Initial analyses with Disulfide-by-design software[18] were negative but manual inspection of the structure revealed a region of the fold that could be amenable to modification if given a simple

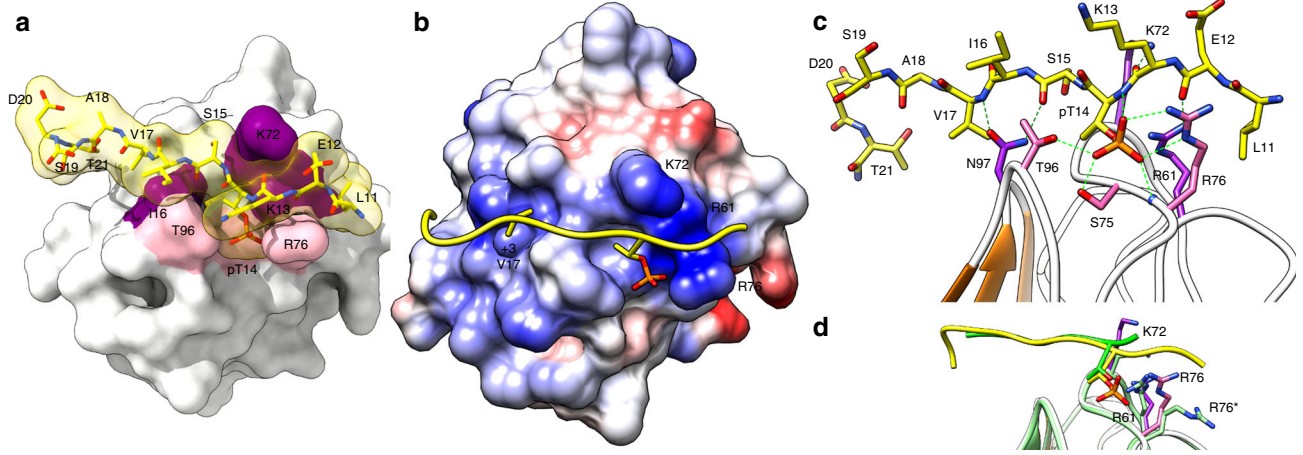

**Fig. 3** Structure of liganded DgcB phosphopeptide complex (FHA domain and exogenous synthetic DgcB phosphopeptide derived from N-terminal tail sequence). **a** Surface representation demonstrating shape complementarity between FHA binding cleft (near side colored pink, far side purple, body of FHA fold white) and phosphopeptide (residues 11–20, yellow/transparent). **b** Charge representation of the peptide-binding cleft (coulombic surface coloring, −10 to +10 kcal/(mol·e), red to blue), with residues R61, K72 and R76 converging at the pThr residue. FHA substrates often use the flanking +3 residue as a determinant of specificity, as demonstrated by V17 projecting into a pocket on the FHA surface. **c** Hydrogen bonding contacts (green dashed lines) between the phosphopeptide and FHA domain, six of which are formed to the phosphoryl group. **d** Structural comparison of the non-phosphorylated (green) and phosphorylated (pink, yellow) DgcB:peptide compexes, demonstrating that the phosphorylated peptide sits relatively deeper in cleft and attracts residues R61, K72 and R76 into conformations that support phosphothreonine recognition (R76 undergoing the largest relative shift, state in non-phosphorylated complex labeled using prime)

rotameric alteration from its observed conformation. The S15 residue immediately after pT14 faces toward a loop present on the FHA β-sandwich that contains S73 (Fig. 4a). In all three of our DgcB structures, the OG of S73 hydrogen bonds to the backbone NH of N124 and sidechain of D71; but manual alteration to a different rotamer would position it towards S15 of the tail. The resulting 2.9 Å distance is longer than the S:S distance in a disulfide, but an S to C substitution at both partners would be isosteric and able to partially flex/move to shorten this gap.

Production of c-di-GMP was monitored by both HPLC analyses and a circular dichroism assay system (modified procedures based on original protocols by Stelitano et al.[19]), alongside the well-characterized YdeH (DgcZ) GGDEF enzyme as a positive control[20]. The R218A mutation at the I-site was engineered into constructs to relieve auto-inhibition and maximize c-di-GMP detection, and a negative control provided via GGDEF catalytic site mutation (E229A, E230A). A basal signal of c-di-GMP was observed in the wild-type enzyme, as a result of co-purification with c-di-GMP (as demonstrated in our full-length structure). Analyses of mutant proteins combining these variations revealed strong activity for only the S15C-S73C-R218A enzyme (Fig. 4b–h), the reaction going to completion ~45 s after GTP addition (at 10 μM protein). In contrast, a single cysteine mutant (S15C-R218A—used to control for the potential effect of cys bringing GGDEFs together in non-specific fashion) displayed no significant activity (Fig. 4c, d, f). We conclude that the above design procedure is valid, and that activation is a specific consequence of FHA cleft occupancy, with the mutant acting as a proxy for phosphorylation. The observation of strong activity for our "disulphide-stapled" mutant led us to investigate whether we could specifically reverse this effect via TCEP-mediated reduction. The CD spectrum shows that post reduction, the double cysteine mutant is still folded (Fig. 4g), and c-di-GMP production is not detectable even at 10 μM protein (Fig. 4h). Taken together, these results demonstrate inducible control of DgcB activity, and further validate our peptide complex structures. Other control experiments confirm that the I-site is not masking wild-type

activity (the single R218A mutant is not active), and that removal of the FHA domain is not sufficient to activate DgcB (the GGDEF-only construct purifies with a small amount of c-di-GMP at the I-site, but does not display significant activity, as ascertained by analysis of the GGDEF-only R218A protein).

**Relation of DgcB oligomeric state to activity.** Stimulated by the observation that DgcB S15C-S73C-R218A licensed cyclic-di-GMP production, we next sought to understand the relationship between DgcB oligomerisation and activity (Fig. 5). Initial size-exclusion chromatography confirmed our structural observation that cyclic-di-GMP holds DgcB in a dimeric state, which converts to a monomer with the R218A substitution (Fig. 5a). In a separate control, the S15C-S73C-R218A variant remains monomeric under reducing conditions (Fig. 5b), but converts to a more complex ensemble when oxidized, which we assigned to five major fractions (Fig. 5c). Calibration to known standards, and validation by non-reducing SDS-PAGE (Fig. 5d), indicate these sizes to be aggregate (peak 1), higher-order oligomer (peak 2), tetramer (peak 3), dimer (peak 4) and monomer (peak 5). The denaturing, non-reducing conditions of the SDS-PAGE indicate that the dimer is almost exclusively disulphide-linked between chains (tail to FHA of opposing subunits) rather than within chains. Assuming that the individual peaks don't interconvert, we assayed these separate fractions for activity, inclusive of controls for total oxidized (all size fractions included) and total reduced protein. Intriguingly, differently sized-oligomers of DgcB all displayed differing activity profiles (Fig. 5e). The reduced and oxidized controls confirmed the earlier finding that the DgcB S15C-S73C-R218A variant could be activated via oxidation. As expected, the non-specific aggregate fraction displayed the least activity of the oxidized samples (peak 1). The dimer (peak 4) was the most active state, displaying faster consumption of GTP than the total oxidized protein (mixed, all oligomers). Monomers, tetramers and other oligomers were all active, but to a lesser degree than the enriched dimeric fraction.

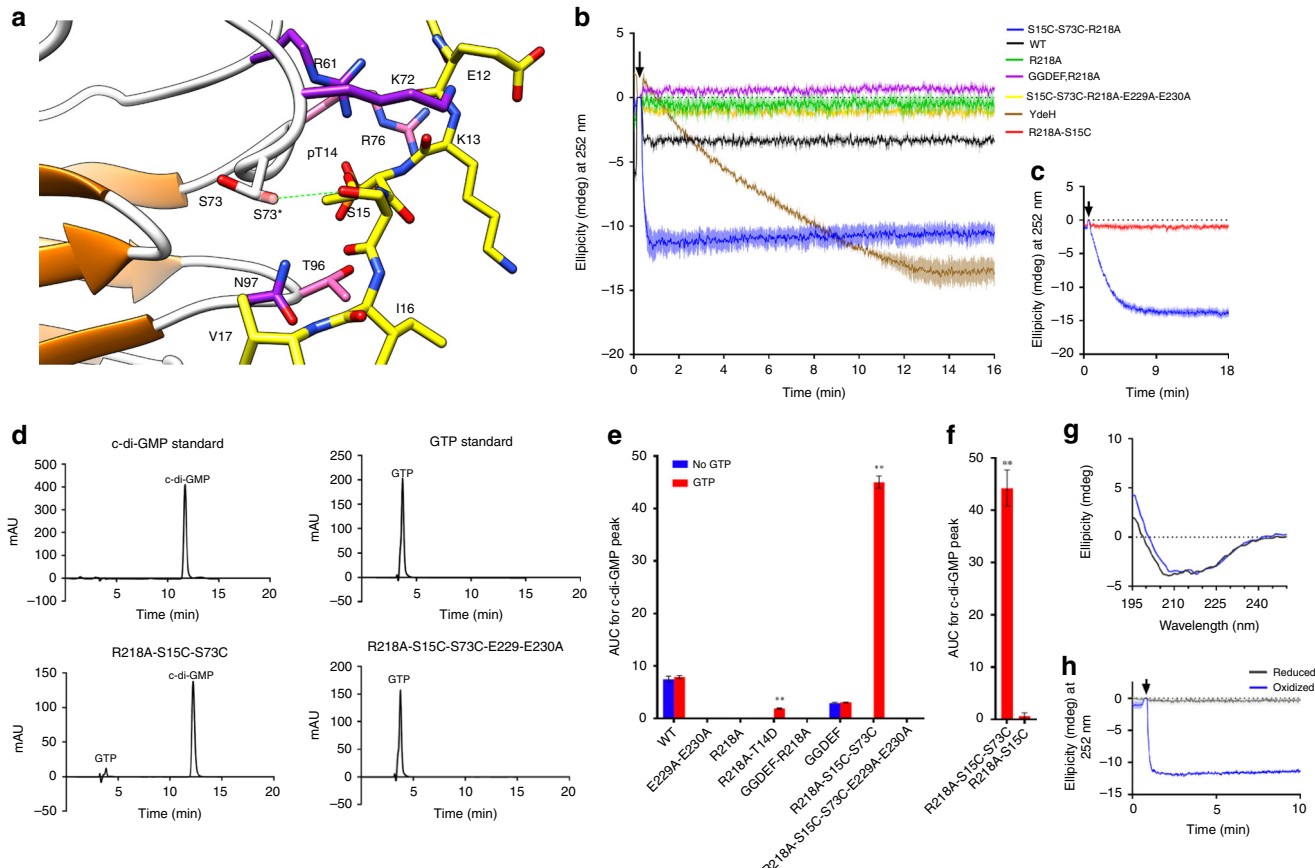

**Fig. 4** Design, validation and regulation of a disulfide-activated DgcB mutant. **a** Design of the disulfide mutation to lock DgcB into a constitutively active state. S15 and S73 were identified as the best candidates for mutation to cysteine residues as to facilitate the formation of a disulphide bond (a different rotamer for S73, labeled with prime, would be in an appropriate orientation to bond S15 when both mutated to Cys). **b** Circular dichroism assay with 10 μM protein. Traces ± standard error. Decreasing ellipticity represents increasing c-di-GMP concentration. **c** Circular dichroism assay with 1 μM of both the single (R218A-S15C only) and double (R218A-S15CS73C) cysteine mutants. The single cysteine mutant has no significant activity compared to the double mutant. **d** HPLC traces of GTP and c-di-GMP standards, and the end products of the enzyme reactions after 40 min incubation. **e** HPLC activity assays monitoring c-di-GMP production at 10 μM protein. Bars ± standard deviation. Significance between samples with and without GTP calculated by Student's $t$-test ($P < 0.001$) through GraphPad prism. **f** HPLC activity assays monitoring c-di-GMP production at 1 μM protein. Significance between R218A-S15C and R218A-S15CS73C calculated by Student's $t$-test ($P < 0.001$) via GraphPad prism. Bars ± standard deviation. **g** Circular dichroism spectra (reporting on protein folding) of both the reduced and oxidized double mutant (at 0.1 μM protein; signal shown is buffer subtracted). **h** Circular dichroism c-di-GMP assay of R218A-S15CS73C mutant using 10 μM protein of reduced and oxidized forms (incubated with and without 12.5 mM TCEP overnight). Source data are provided as a Source Data file

## Discussion

We have structurally determined several states of the *Bdellovibrio* diguanylate cyclase DgcB, allowing us to document the mechanisms behind both inhibition and activation of c-di-GMP production. The full-length form of DgcB is informative of regulation, providing information on several domain interfaces (FHA:FHA, FHA:GGDEF and GGDEF:GGDEF). The conformation we observe likely represents a feedback-inhibited state wherein the GGDEF domain has been locked by the c-di-GMP product into a non-productive conformation; we posit that this represents a form present after entry of prey. This state is further stabilized/inhibited by protein:protein interactions—the FHA domains making contacts to one another, and also the GGDEF domains (in particular chain A situated between the two α0 helices). DgcB binds c-di-GMP using both I-sites of the GGDEF dimer, forming a pseudosymmetric protein:nucleotide complex. This mode of inhibition is distinct from most well-characterized GGDEF proteins, which usually complex c-di-GMP via a single I-site in conjuction with another region of the fold (e.g. the REC

domain of PleD[12]). To the best of our knowledge, this two-fold arrangement has only been observed in the Dcsbis GGDEF from *P. aeruginosa*[21] where the enzymatic domain is appended to a differing GAF sensory domain. This observation is related to the asymmetric orientation of the stimulus-sensing FHA domains, i.e., a symmetric GGDEF dimer centred around the I-site constricts the space available at the N-terminal end of the fold. Hence, most GGDEF proteins do not adopt this orientation as it would lead to steric clashes of the appended sensory domains, but in DgcB this conformation is enabled by having the chain B FHA domain offset. Supportive of this idea is the finding that the I-site symmetrical dimer of Dcsbis was obtained from a truncated construct (lacking the sensory GAF domain), and that the full-length protein adopts a different conformation.

FHA asymmetric intercalation into the GGDEF dimer appears to be relevant because the hydrophobic residues involved are conserved in DgcB homologues, and both the FHA:FHA and GGDEF:GGDEF orientations compatible with this arrangement have been observed independently in multiple structures (FHA

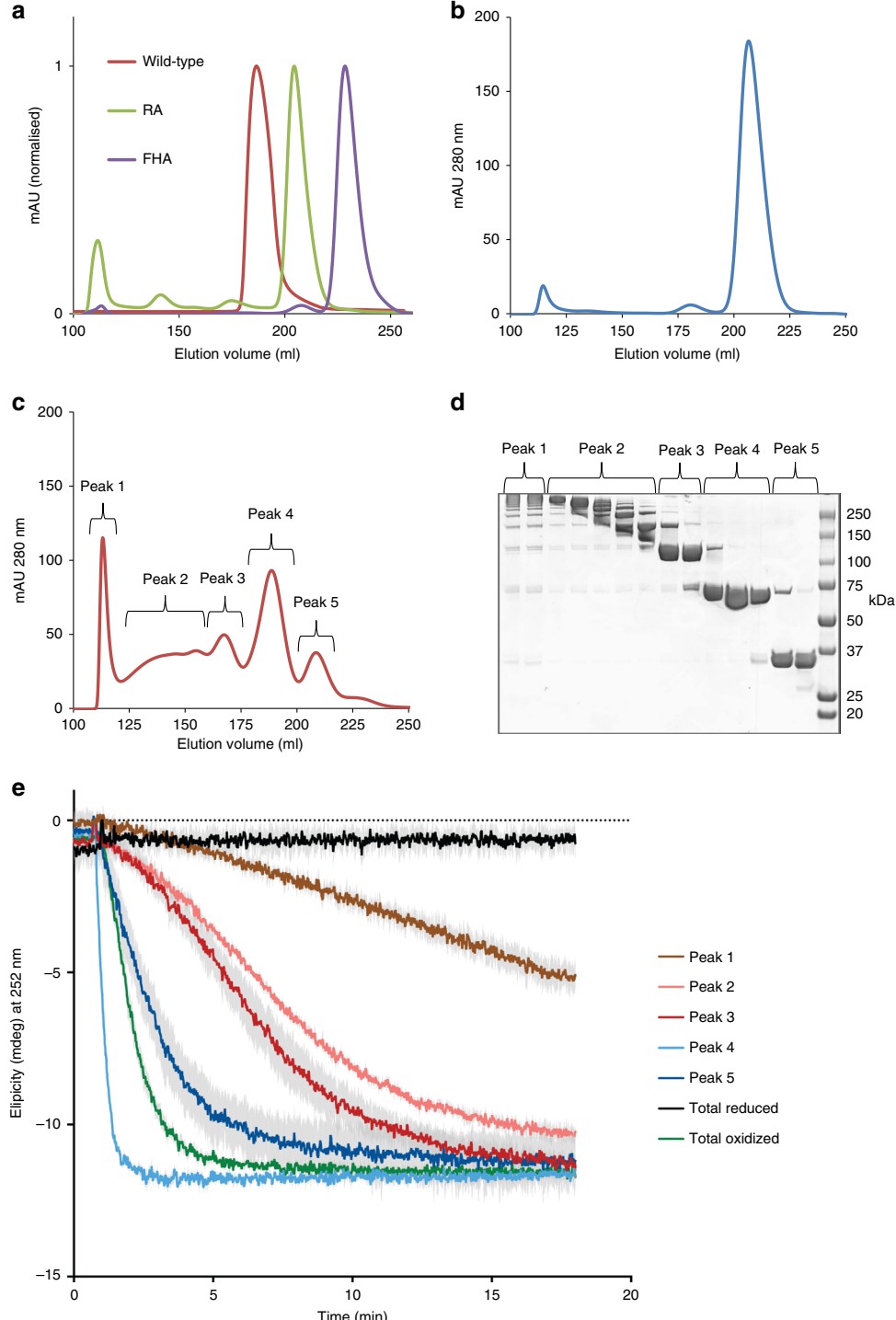

**Fig. 5** The effects of DgcB oligomerization state on diguanylate cyclase activity. **a** Purified DgcB wild-type, R218A mutant and FHA domain proteins were analysed by gel-filtration and hypothetical molecular weights were estimated by comparison with the elution profiles of proteins of known molecular weights. **b** Gel-filtration profile of DgcB S15C-S73C-R218A mutant protein under reducing conditions and **c** under non-reducing conditions following incubation with oxidised glutathione to drive disulfide bond formation. Elution fractions collected during this experiment were assigned to five different peaks as noted on the trace. **d** Non-reducing SDS-PAGE analysis of fractions of DgcB S15C-S73C-R218A collected from gel-filtration experiment of oxidised glutathione treated protein. The five different peaks were pooled, concentrated and used in CD assays with GTP to monitor the diguanylate cyclase activity of these different oligomeric species. **e** CD spectra demonstrating activities of the five eluted peaks of DgcB S15C-S73C-R218A. Samples of the oxidised pool (total oxidized, all oligomers) prior to gel-filtration and reduced protein (total reduced, all oligomers) were also included as controls. Source data are provided as a Source Data file

the three separate forms of this study, GGDEF homologues in other studies[1]). Asymmetry can be a powerful means of regulating signaling networks, *e.g.* as observed in the ChpT-CtrA phosphorelay of *Brucella abortus*[17]. Like other GGDEFs, I-site inhibition of DgcB is dominant over the active state, as shown by our activity assays (Fig. 4). Physiologically, this mechanism exists to set an upper limit for local [c-di-GMP] at the prey-interacting pole of the *B. bacteriovorus* cell, which from DgcB production would persist over a very specific (relatively short) timeframe during prey invasion[8].

The orientation of the GGDEF domains in an active enzyme is presumed to be 180˚ from that of the I-site inhibited form—this productive state would allow the GG(D/E)EF motifs (of a different dimeric arrangement) to approach one another, leading to two molecules of GTP positioned correctly to form the cyclic product. It is important to consider the DgcB structure in the context of general features of GGDEF activation - almost without exception, GGDEF proteins with differing stimulus-sensing/input domains (Rec, PAS, GAF, HAMP, transmembrane, usually N-terminal to the enzymatic domain) use a helical coiled-coil (α-1) to communicate the regulatory signal[22]. When comparing five well-characterized GGDEFs, this region demonstrates relative flexation, leading to a generalized model that uses the tilt angle between α-1 and the catalytic domain to modulate the frequency of GGDEF:GGDEF encounter and thus c-di-GMP formation[22]. The tilt/communication may be regulated by two continuous features – a heptad repeat region of the coiled coil, and a consensus DxLTxxxN/SR/K motif that forms a wide β-turn that leads into the GGDEF fold. The differing conformation of the $D^{150}xLTxxxSK^{158}$ regions in DgcB is of interest—chain B places this motif in the consensus β-turn conformation found in the majority of GGDEF structures. In contrast, the DxLT component of chain A forms part of a helix in the interdomain linker, unwinding the region afterward such that S157 projects toward D221 of the I-site RxxD motif. There is no suggestion that this unusual conformation of the chain A DxLT motif is mechanistically relevant—the small helix (aa 147–154) would be peripherally located/isolated in a GGDEF dimer where the active sites were productively engaged. However, the plastic nature of this region in general is of interest given the aforementioned tilt model of activation and requirement for multiple conformations; our structures suggest that the arrangement of the DxLT motif can alter into a stable form, rather than an order/disorder transition.

A substrate complex representative of a competent catalytic state has yet to be obtained for a GGDEF enzyme, the closest matches being a post-catalysis form of YdeH/DgcZ[23] (with c-di-GMP product spanning two active sites), a complex of *Is*PadC (GTP locking the monomers together rather than juxtaposing active sites)[24] and an active/empty *Is*PadC heterodimer with quasi-translated coiled coils[25]. GGDEF activation can be triggered by stimulus-induced dimerization (e.g. PleD[12]) and/or stimulus-induced domain rearrangement (often relieving self-inhibitory interactions, e.g. DosC[26]). The observation of the DgcB FHA sensory domain in a common dimeric conformation across our three separate structure determinations suggests that this complex is stable (as a constitutive dimer, even in the absence of the GGDEF domain), and would need at least partial rearrangement to place the GGDEF domains into an active conformation.

The orientation of the DgcB FHA domains in a form lacking c-di-GMP inhibition remains enigmatic, and we were unsuccessful despite extensive trials with a R218A mutant; if the FHA domains kept the same interaction as the three structures presented herein, one FHA phosphopeptide cleft would be exposed and the other sequestered (via R61 of the "closed" monomer binding to E64 of the "open" monomer, and D56 inserting into the "closed" site

near to where the phosphothreonine binds). Our structure of the FHA:phosphopeptide complex demonstrates that cleft occupancy does not alter the FHA:FHA contacts, at least for a peptide added in trans. Hence, rearrangement will presumably require cis phosphopeptide binding, wherein the two elements that flank the pThr epitope (aa 1–10, 22–31) can alter the surface properties of the FHA domain and change the domain:domain packing; tail-binding could promote new interactions (e.g. in the inter-domain linker region) and/or relieve inhibitory interactions (e.g. our observed FHA:GGDEF interface). Our observation that both the unphosphorylated tail (intrinsic) and phosphorylated form (extrinsic) can bind the DgcB FHA domain is in keeping with other (non-GGDEF) FHA proteins, e.g. GarA[15], and the mode of peptide binding is similar despite the pronounced differences in FHA sequence and functional usage. Our activity assays demonstrate that this "non-specific" binding does not appear to result in any significant c-di-GMP production (Fig. 4b, e), presumably providing tight control on the process of prey invasion. We can conclude that the I-site c-di-GMP is self-produced, because the E229-E230 variant does not co-purify with nucleotide (Fig. 4e); so the small amount of wild-type basal activity is enough to eventually saturate the protein over the longer timescale of recombinant production.

From careful searching of the literature, we are not aware of any other covalently-locked FHA variants; this methodology may thus be of interest as a tool to study other signaling systems. The vastly increased c-di-GMP production of the S15C-S73C-R218A mutant of DgcB can be interpreted (as intended from design guided by the phosphopeptide complex) as a locked active form that has lost I-site inhibition. Historically, it has been easier to design obligate "off" GGDEF proteins versus obligate "on" variants, with a notable exception being a GCN4 zipper:GGDEF fusion protein with high activity that dimerizes around an introduced strong coiled-coil forming segment[27]. The differences between wild-type DgcB (no appreciable activity), T14D-R218A (phosphomimetic, minor c-di-GMP production) and S15C-S73C-R218A (presumed locked tail, strong activity) are very suggestive of DgcB activation via FHA tail phosphorylation. This hypothesis is strongly supported by the ability to reverse the activation of the S15C-S73C-R218A form using TCEP to reduce the disulfide between the tail and FHA domain (Fig. 4g).

There are several potential modes in which tail-driven reorganization of the observed asymmetric FHA:FHA interface could occur, all of which have been observed to operate for different FHA-containing proteins: formation of intradomain contacts (tail binding self, akin to GarA[15], OdhI[28]) or interdomain contacts (tail binding oligomeric partner, head-to-head, head-to-tail, or as a fibre akin to TIFA[29]). Head-to-tail and fibre models would require an active form larger than a dimer to place the GGDEF domains together. Our dissection of the differing activities for different oligomeric states of DgcB (Fig. 5) suggest that although monomers, tetramers and higher-order oligomers all demonstrate some activity, the most-highly active species was the dimeric form. We are able to combine intra- and head-to-head inter-domain models with both our observed structures and general principles for GGDEF activation into a working model for the DgcB active state (Fig. 6). There is some evidence from our reducing SDS-PAGE that the inter- linked conformation (tail into opposing monomer) may be more prevalent in the active dimers (Fig. 4d). Herein we have used constraints from a dimeric pseudoactive *Is*PadC structure (5LLW[24]) to position two copies of chain B from full-length DgcB. The resultant orientation is sterically compatible with both FHA domains and bound phosphopeptide tails (superimposed), leaving the linker peptide appropriately positioned to span the region occupied by the *Is*PadC coiled coil. Adoption of this state would require both the

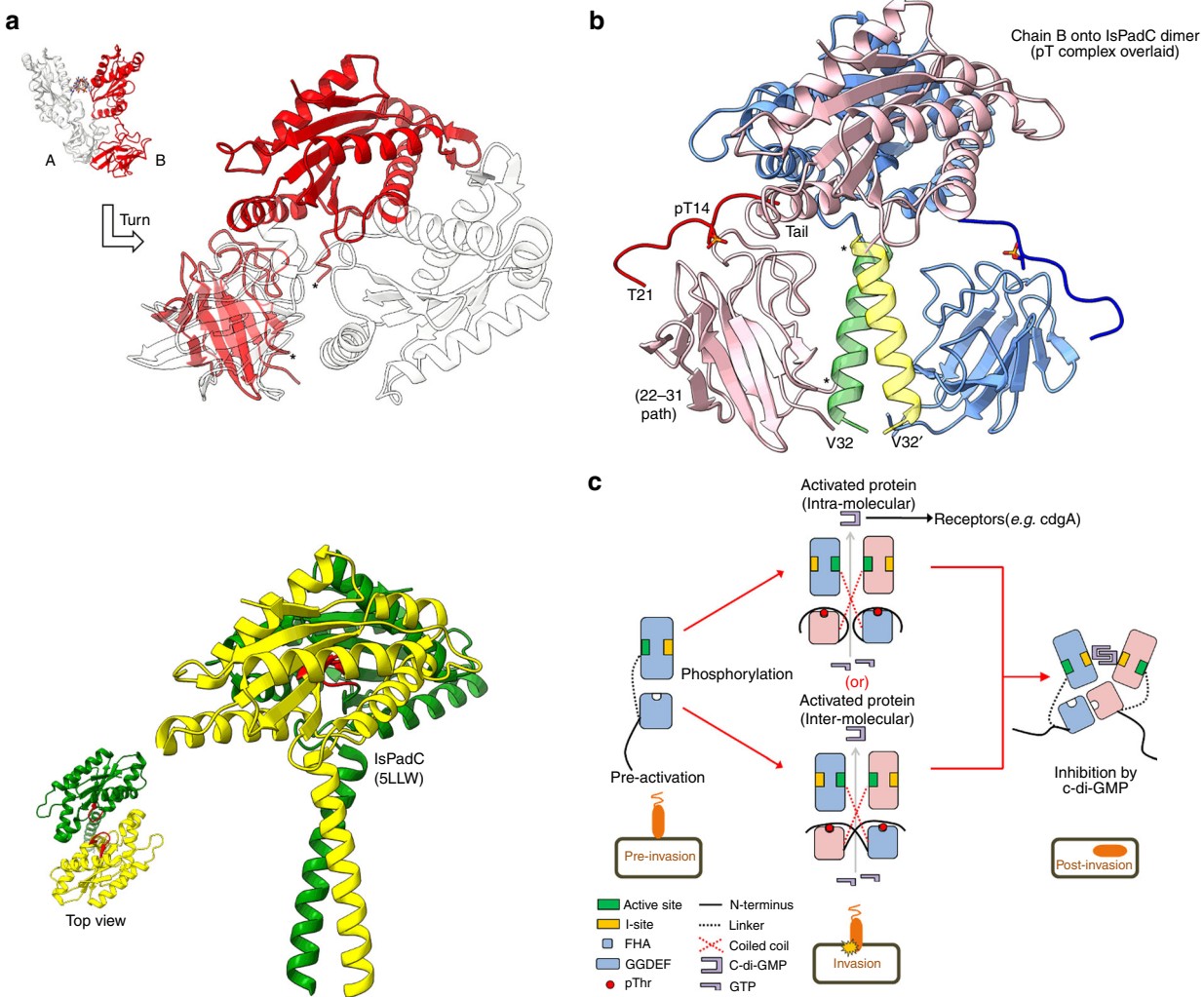

**Fig. 6** Proposed model for DgcB activation. **a** Inset, DgcB representation as oriented in Fig. 1. This same representation (chain A white, transparent; chain B red) is re-oriented in main panel to demonstrate that the active state model in **b** is derived from chain B of our experimentally determined full-length structure. Lower panel, near-active GGDEF dimer of *Is*PadC (PDB 5LLW, sensory domains removed, individual chains yellow and green), displaying a conformation of coiled coils necessary to juxtapose the two active sites (red loop, top view). **b**) Proposed model of active DgcB generated using composite structures. Chain B of DgcB (FHA and GGDEF, pink) was overlaid onto *Is*PadC dimer (generating a second chain of DgcB, colored blue). The activating coiled coil region of the *Is*PadC dimer is shown (green, yellow), situated between the FHA and GGDEF domains of DgcB (the termini of which are labeled with an asterisk, and presumed to lead into a similar coiled coil in the active form). The N-terminal phosphopeptide ligand from our complex structure is superimposed and shown in bold, with pT14 in stick form. T21 of the N-terminal tail and V32 of the FHA domain are labeled to demonstrate the likely path of the (missing) 10 linker residues. **c** Schematic for DgcB conformational states pre-, during and post-activation. Phosphorylation of residue T14 (from prey encounter-derived activation of a kinase) leads to either an intra- (top) or inter- (bottom) FHA:tail-driven GGDEF dimer (the model in **b** allows both and cannot be used to suggest which is more likely; non-reducing SDS-PAGE results of Fig. 5 suggest inter-dominates). This rearrangement results in productive association of the GGDEF active sites (dark green) and synthesis of c-di-GMP. The resultant c-di-GMP signals downstream invasion processes until an upper threshold is achieved and DgcB subsequently inhibited via I-site mediated feedback (I-site, orange block; as observed in our full-length structure). The *Bdellovibrio bacteriovorus* lifecycle stages corresponding to these events are represented below (predator orange, prey dark brown)

linker (aa 135–143) and start of wide turn (aa 144–148) to adopt a helical conformation (~2 heptad repeats), which is interesting given our observation of a small helix preceding the turn in chain A of the full-length DgcB. The distances involved are similar for either intra- or inter- FHA:tail interactions. For the purposes of tail-binding driving conversion into the active state, this model suggests that the two flanking regions (either side of the tail) could assist by orienting away from the coiled coil space (aa 22–31) and possibly contacting the GGDEF domains (aa 1–10). Tail-binding would also occupy the FHA cleft, relieving the FHA: FHA contacts observed in all three of the structures presented here. Further, our model uses a 1:1 stoichiometry of tail:FHA, as

we believe the 1:2 ratio of our phosphopeptide complex structure to be related to crystallization conditions. Lastly, the FHA: GGDEF contacts observed in our full-length structure are incompatible with an occupied FHA cleft, suggesting that part of DgcB activation could result from GGDEF "release" from this state and associated increase in conformational freedom.

Our confirmation herein of phosphopeptide-gating of DgcB activity, ultimately licensing *Bdellovibrio* local cyclic-di-GMP production and therefore predation, requires context regarding upstream and downstream signaling events. The kinase responsible for phosphorylating DgcB T14 has not been identified (and this is beyond the scope of this study), but presumably senses

prey encounter and has multiple protein substrates. Our working hypothesis is that the kinase would sense a stimulus upon prey encounter (Myxococci sense osmolarity changes when exposed to prey[30]), and become active, phosphorylating T14 of DgcB to stimulate conformational change into a state capable of producing cyclic-di-GMP. Supporting this hypothesis, the distantly related predator *Sorangium cellulosum* encodes for a putative 974aa fused kinase-tail-FHA-GGDEF protein (UniProt BE21_29150), whose kinase domain is most similar to *Bdellovibrio* gene product Bd3148 (26% identity; interestingly this also shares 27% identity with the PknG kinase that phosphorylates the OdhI FHA tail[31]). Intriguingly, Bd3148 is most-highly expressed during attack phase when *Bdellovibrio* is ready to initiate predation[32] and has been identified as compromising predation when interrupted by transposon insertion[33]. The dominance of I-site inhibition questions whether a phosphatase is needed to limit DgcB activity, but in support of such a hypothesis, a putative phosphatase exists in the same operon as *dgcb/bd0742* (bd0740, PFAM PF10049 calcineurin-like family). Downstream of DgcB activation, CdgA (Bd3125) is a c-di-GMP receptor with an invasion-related phenotype, organizing processes at the "biting" pole of *Bdellovibrio*[8]; the stronger phenotype of ΔDgcB over ΔCdgA is suggestive of more than one target for DgcB-produced c-di-GMP.

Our investigations have provided the first details behind the mechanisms used to activate DgcB function, and thus signal prey invasion in the bacterial predator *Bdellovibrio bacteriovorus*. We provide the molecular basis for initiation of c-di-GMP production, demonstrating that the stimulus is a phosphopeptide tail present on the DgcB GGDEF synthase itself. Our model allows the design of a sensor-locked, obligate "on" variant, whose activity is validated by *in-vitro* assays; this mode of distant stimulation mimics the natural stimulus (phosphopeptide occupancy) and is distinct from other studies that alter the GGDEF coiled coil region directly to achieve activation[24,27]. These observations provide a starting point for exploring the signaling network that underpins entry into the staged lifecycle of this bacterial predator and further enrich our general understanding of GGDEF activation/regulation.

## Methods

**Cloning**. [NB the original HD100 gene annotation for DgcB has the start sequence misannotated, mature protein starting at M[11]AHN of UNIPROT entry Q6MPU8. Throughout this manuscript the DgcB numbering follows our start site annotation.] Primer sequences are provided in Supplementary Table 1. Full-length (aa 1–310), GGDEF domain only (aa 148–310), FHA domain with N-terminal tail (aa 1–135), and FHA domain only (aa 33–135) constructs of *dgcb* were amplified from *B. bacteriovorus* HD100 genomic DNA. The amplified genes were inserted into a modified pET41 expression plasmid (Novagen, thrombin cleavable 8xHis-tag introduced at C-terminus, and GST-tag removed) by restriction-free cloning. Mutant variants (T14D, S15C, S73C, R218A, E229A, and E230A) of DgcB constructs were produced by standard site-directed mutagenesis. Constructs were confirmed by sequencing, and introduced into the *E. coli* expression strain BL21λDE3 (New England Biolabs). A pET28a plasmid encoding *ydeH* (kindly provided by the Schirmer laboratory, constructed by Dr. Alexander Böehm, Biozentrum Basel) was transformed into BL21λDE3.

**Protein production and purification**. Cells containing DgcB constructs were grown at 37 °C (shaking incubation at 180 RPM) in 2× LB supplemented with 100 μg/ml kanamycin until an $OD_{600}$ of 0.6–1 was reached. Gene expression was then induced with 0.5 mM IPTG and cultures were grown overnight at 15 °C. Cells were harvested by centrifugation at 6675 × *g* for 6 min and frozen at −20 °C (media supernatant discarded). Cells were resuspended in buffer A (20 mM imidazole pH 7.0, 400 mM NaCl, and 0.05% v/v Tween 20) and lysozyme by gentle shaking at 10 °C. Cells were lysed on ice by sonication and centrifuged at 48,400 × *g* (4 °C) for 1 h to clarify lysate. The supernatant was then loaded onto HisTrap FF nickel columns (GE Healthcare) pre-equilibrated in buffer A. Columns were washed with 12 column volumes of buffer A. Two 20 ml elution steps at 8% buffer B (400 mM imidazole pH 7.0, 400 mM NaCl and 0.05% v/v Tween 20) in buffer A and 100 % buffer B were used to elute protein from the column. The protein eluate was dialyzed overnight at 4 °C. Proteins were concentrated by Vivaspin® spin-

concentrators (Sartorius) and purity confirmed by SDS-PAGE. For the RP-HPLC and CD assays, proteins were dialyzed into 20 mM HEPES pH 7.0 and 200 mM NaCl. For crystallization, final dialysis buffers were composed of - wild type full-length DgcB: 20 mM HEPES, 200 mM NaCl and 2 mM β-mercaptoethanol; FHA domain N-terminal tail construct: 25 mM Na citrate pH 5.0, 100 mM NaCl and 10 mM HEPES pH 7.0; FHA only construct: 20 mM HEPES pH 7.0 and 200 mM NaCl.

BL21λDE3 with the YdeH construct was grown at 37 °C (shaking incubation at 180 RPM) in AI media (total volume 1 litre in dH₂O: 10 g tryptone, 5 g yeast extract, 3.55 g Na₂PO₄, 3.4 g KH₂PO₄ and 2.68 g NH₄Cl, 2.5 ml of trace metals, 0.1 g glucose, 0.4 g lactose, 1 ml glycerol and 2 mM MgSO₄). At an $OD_{600}$ of 0.6–1, cells were transferred to a 15 °C shaking incubator for O/N expression. Cells were harvested by centrifugation at 6675 × *g* for 6 min. YdeH was purified from a method adapted from that published previously[20]. Cells were lysed by sonication in buffer C (50 mM NaH₂PO₄ pH 7.5, 200 mM NaCl, 10 mM imidazole, 50 mM L-glutamic acid and 50 mM L-arginine) and lysozyme before clarification (as performed for the DgcB constructs). Lysate was loaded onto HisTrap FF nickel columns (GE Healthcare) pre-equilibrated in buffer C and eluted with a stepwise gradient of buffer C supplemented with imidazole (up to 400 mM). Fractions were dialyzed into buffer D (20 mM TRIS pH 7.6, 150 mM NaCl, 50 mM L-glutamic acid and 50 mM L-arginine) overnight at 4 °C. Purity was confirmed by SDS-PAGE.

**Reducing size-exclusion chromatography for assay samples**. DgcB variants (WT, FHA, R218A, S15C-S73C-R218A) were purified by nickel affinity chromatography as described above and then dialysed into reducing gel-filtration buffer (20 mM tri-sodium citrate, pH 6.5; 250 mM NaCl; 2 mM DTT) prior to concentration by ultrafiltration to 50–80 mg/ml. Proteins were then subject to gel-filtration on a Superdex 200 26/60 column (GE Healthcare) equilibrated in the dialysis buffer.

**Non-reducing chromatography & SDS-PAGE of oxidized samples**. For analysis of disulfide-induced protein oligomerisation, DgcB S15C-S73C-R218A protein was prepared by gel-filtration as described above and then fractions of pure protein were pooled, concentrated to 10 mg/ml and dialysed extensively against gel-filtration buffer lacking DTT (non-reducing gel-filtration buffer). Protein was then supplemented with 5 mM oxidised glutathione and incubated overnight with gentle agitation. This sample was further concentrated to 50 mg/ml before being subject to further gel-filtration with non-reducing gel-filtration buffer. Fractions of eluted protein were collected and analysed by non-reducing SDS-PAGE to assess disulfide induced cross-linking. The elution profile of DgcB S15C-S73C-R218A under non-reducing conditions and the non-reducing SDS-PAGE gel were used to subdivide the elution profile into five distinct peaks, which were then concentrated to >10 mg/ml for use in CD assays.

**Structure determination**. Crystals were grown at 18 °C using the sitting drop technique. For the FHA domain:phosphopeptide complex (aa 33–135), protein was mixed at a 1:1 stoichiometry with a C-terminally amidated phosphopeptide with the amino acid sequence NSDNLEK(p)TSIVASDT (where (p)T is a phosphorylated Thr residue). Crystals were obtained in 0.2 M K bromide, 0.1 M Na acetate pH 5.5 and 15% w/v polyethylene glycol (PEG) 4000 (for aa 1–310, 40 mg/ml protein); 0.2 M ammonium formate, 10% w/v polyvinylpyrrolidone and 20 % w/v PEG 8000 (for aa 1–135; 5 mg/ml protein), and 0.1 M Na-HEPES pH 7.0 and 20 % w/v PEG 8000 (for aa 33–135, 5 mg/ml protein). Crystals were cryoprotected in mother liquor supplemented with 20 % (v/v) ethylene glycol before being flash cooled in liquid nitrogen. Diffraction data were collected at the Diamond Light Source in Oxford, UK. Data reduction and processing was completed using iMosflm[34] (aa 1–310) or the xia2 suite[35] (aa 1–310; aa 33–135). Molecular replacement of the full length structure (aa 1–310) used both FHA and GGDEF search model ensembles of PDB codes 3IGN, 3TVK, 2WB4, 3I5B, 3ICL, 4QCJ, 2XT9, 3OUN, and 3PO8 via PHASER[36]. A clear solution was obtained with two copies of DgcB in the asymmetric unit. The FHA domain of the refined full length structure was used to solve both the aa 1–310 form (Four copies in the AU) and aa 33–135 form (two copies in AU). Protein structures were built/modified using COOT[37], with cycles of refinement in PHENIX[38] and PDB-REDO[39].

**Reverse phase high-performance liquid chromatography assay**. Two reaction mixes (total volume in each, 100 μl) composed of 10 μM protein, 20 mM HEPES pH 7.0. 200 mM NaCl and 10 mM MgCl₂ (with one of the reaction mixes containing 100 μM GTP) were set up for each protein assayed. To compare the single to the double cysteine mutant 1 μM of protein was used. Reaction mixes were incubated at 24 °C for 40 min. The reaction was heat inactivated at 98 °C for 10 min. Samples were clarified by centrifugation at 13,000 RPM for 15 min in a benchtop centrifuge. The supernatant was centrifuged for an additional 20 min at 13,000 RPM before being gently transferred into HPLC micro-sampling vials. Samples (4 μl) were analyzed on a kinetex 1.7 μm (particle size) C18 100 Å (pore size), 150 × 2.1 mm column connected to a Dionex Ultimate3000 UHPLC system with a detection wavelength of 252 nm. The flow rate was set at 0.1 ml/min. The mobile phase was 100 mM Na phosphate pH 5.8/methanol (95/5, v/v). Three technical repeats were undertaken for each protein sample.

**Circular dichroism assay**. Reaction mixes (total volume 3 ml) consisting of 10 μM protein, 200 mM NaCl, 20 mM HEPES pH 7.0 and 10 mM MgCl₂ were set up in a 1 cm pathlength quartz cuvette (For YdeH, buffer D supplemented with 10 mM MgCl₂ was used). For size excluded samples a modified buffer of 20 mM tri-sodium citrate, pH 6.5, 250 mM NaCl, and 10 mM MgCl₂ was used to prevent aggregation of the mutant protein. Additionally, the single and double cysteine mutants were monitored at a protein concentration of 1 μM. Circular dichroism (CD) measurements were taken in a Jasco J-1500 instrument with a PTC-517 accessory at a constant 25 °C. Sample was continuously mixed within the cuvette by a magnetic stirrer set at 100 RPM. The CD detection wavelength was set at 282 nm. Reaction was initiated by GTP (at a final concentration of 100 μM) and monitored every second. A total of three technical repeats were performed for each protein sample. The effect of reduction was tested by overnight incubation of the double cysteine mutant (specifically S15C-S73C-R218A) in reaction buffer supplemented with 12.5 mM TCEP. To confirm retention of ordered structure, this sample was diluted 1 in 100 in dH₂O and a regular CD spectrum was taken from 195 to 250 nm.

**Reporting summary**. Further information on research design is available in the Nature Research Reporting Summary linked to this article.

### Data availability

Coordinates and structure factors have been deposited in the PDB under accession codes 6HBZ for the full-length protein, 6HC0 for the FHA-only structure, and 6HC1 for the FHA:phosphopeptide complex. The source data underlying Figs. 4 and 5 are provided as a Source Data file. Other data are available from the corresponding author upon reasonable request.

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

### Acknowledgements

We would like to thank the Sockett lab (Nottingham) for instigating the project and their support/advice throughout. We are grateful to Charles Moore-Kelly for assistance with the CD assays. R.W.M. was supported by a BBSRC MIBTP studentship. This research was funded by the U.S. Army Research Office and the Defense Advanced Research Projects Agency accomplished under Cooperative Agreement Number W911NF-15-2-0028. The views and conclusions contained in this document are those of the authors and should not be interpreted as representing the official policies, either expressed or implied, of the Army Research Office, DARPA, or the U.S. Government. The U.S. Government is authorized to reproduce and distribute reprints for Government purposes not withstanding any copyright notation hereon. P.J.M. wishes to acknowledge support in the form of a Future Leader Fellowship and a David Phillips Fellowship from the UK Biotechnology and Biological Sciences Research Council (BB/N011945/1 and BB/S010122/1).

### Author contributions

R.W.M., P.J.M., I.T.C. and A.L.L. designed the experiments; R.W.M., I.T.C. and P.J.M. performed the experiments; and R.W.M., I.T.C. and A.L.L. interpreted the results and wrote the manuscript.

## Additional information

**Competing interests:** The authors declare no competing interests.

