## [Peer Review File · Nature Communications]

Reviewers' comments:

Reviewer #1 (Remarks to the Author):

The manuscript entitled «Structural basis for activation of *Bdellovibrio* diguanylate cyclase that licenses prey entry» by Meek and co-workers, describes three interesting crystal structures of the cyclic-di-GMP synthase DgcB at high resolution. The analysis of the full-length and FHA-only structures enabled the determination of the FHA-only/phosphopeptide structure and ultimately the design of a constitutively active “disulfide-stapled” mutant comprising an engineered disulfide bridge. The work is technically sound and the manuscript is well written, albeit my personal impression is that the discussion section could be streamlined.

Although I find the work very interesting, particularly the resolutions are significantly better than the deposited structures of DgcB homologues, I am not excited about it. Particularly, I ask myself what do we really learn about the activation mechanism of this interesting family of enzymes (as the authors imply in the title)? The full-length structure discloses little more than the structures of the isolated domains, because even the domain connectivity relies on distance constraints due to the lack of continuous electron density. The FHA-only structures confirm recognition of the phosphorylated N-terminus. For the *Bdellovibrio* enzyme, this is indeed new, but it is well-known that FHA domains recognize pThr residues as stated in reference #8 and shown in figure 2d. Thus, the manuscript describes a solid piece of work, which agrees well with the current state of knowledge in the field, which is good, but after reading the title, I would expect at least some solid data on how peptide binding activates the catalytic domain.

The observation that the FHA:FHA interface seen in the full-length structure is conserved in the FHA-only structure (and perhaps in the FHA:phosphopeptide structure as well) is an interesting observation, particularly because the pocket is partially occupied by a bound ion. Can the authors say what kind of ion it is? Is this interaction biological relevant? Do the constructs form oligomers in solution? I would suggest to include a superposition of the FHA:FHA interface seen in all three structures, because I only learned from the discussion (page 16, line 377) that the contact is the same in all three structures. If this contact has any biological relevance, mutations of interface residues should alter the catalytic activity. In figure 5c the authors assume that the biologically active DgcB species is a dimer and that the N-terminus can only bind intra- or inter-molecular. Is a daisy chaining event with subsequent oligomerization via FHA:N-terminus interactions unlikely? The same question arises for the hyperactive “disulfide-stapled” mutant. In principle, the presumed disulfide bridge could be intra- or inter-molecular. I would expect that both species have different activities and different retention times in size-exclusion chromatography. If there is only one dimeric species of this mutant a non-reducing SDS gel should reveal if the disulfide bridge is intra- or intermolecular. Generally, I have the feeling that some sensible solution data would be very helpful for the proper analysis of these interesting crystal structures.

Reviewer #2 (Remarks to the Author):

Bdellovibrio bacteriovorus is an organism that consumes other bacterial cells in a complex lifecycle that involves the bacterial second messenger c-di-GMP. *Bdellovibrio* encodes several enzymes that make and break c-di-GMP in addition to an abundance of potential c-di-GMP receptors. While genetic and cellular studies have assigned distinct functions of many of the enzymes involved in c-di-GMP signaling, the mechanistic underpinnings and in particular the regulation of these enzymes remained less well defined. The manuscript by Meek et al. describes a thorough structure-function analysis of one such enzyme, a unique diguanylate cyclase (DGC) of *Bdellovibrio*. The protein, DgcB, is comprised of a sensory FHA domain flanked by an N-terminal tail and a C-terminal GGDEF domain. Based on three novel, insightful crystal structures (full-lengths, isolated FHA domain, and isolated FHA domain bound to phosphopeptide), the authors derive the central hypothesis that the N-terminal peptide of DgcB itself becomes the activating signal of the FHA domain when phosphorylated. They go on to test this model by structure-informed engineering of a hyper-active enzyme. The study is rigorous and provides interesting new data and models to the c-di-GMP field, and more generally, to the regulation of *Bdellovibrio*'s fascinating biology.

There are only a few minor points the authors should consider when revising the manuscript. Many of the points can be addressed by editing the text. Some key experiments would add value to the current study; they should be feasible, making use of available reagents and the expertise of the investigators:

1. Introduction: The first paragraph could use some key references to primary literature. While the review cited in the first sentence covers most of what is described, explicit mentioning of key publications would be a welcome addition.
2. Introduction, line 62: Is there is a reason why the number of PilZ proteins cannot be determined precisely?
3. Throughout the manuscript but particularly in the introduction: The excessive use of texts in brackets has a somewhat negative impact on readability.
4. Figures (all): Please increase the label font size in the figure panels. Many are hard to read in a printed version of the figures.
5. Line 224: The authors describe the FHA:tail interaction as occurring "in-cis". In the most stringent meaning, this would imply that the peptide tail comes from the same polypeptide chain as the FHA domain. Based on the previous structural description in the manuscript, this argument cannot be made conclusively. To stay on the safe side, I would rephrase this sentence so that it includes the possibility of the tail reaching over from an adjacent DgcB molecule (in a dimeric/multimeric assembly).
6. Line 231: The authors only describe phosphopeptide binding to chain B of the complex structure. How does chain A compare? If chain A lacks clear indications of peptide binding, please comment on the possible reasons for non-equivalent protomers and whether you think that such a difference could have mechanistic implications.
7. Figure 3: Have the authors attempted to determine the affinity and stoichiometry of phosphorylated vs unphosphorylated tail peptide binding to the FHA domain? This is only a minor point since the interaction could occur in cis or within a complex, and as a result the affinities of free peptide may only approximate the interaction strength within the full-length protein.
8. Figure 4: In an elegant approach, the authors engineer a disulfide bond between FHA domain and the N-terminal tail that vastly increases DGC activity of the enzyme. Have the authors tried to add phosphorylated peptide in trans to full-length of N-terminally truncated DgcB? Such an experiment appears feasible and would add to the mechanistic studies as it could present activation by the native ligand. Similarly, the TCEP reduction experiment is neat as it shows reversibility of the disulfide bond effect. I wonder if the authors have attempted to re-oxidize the cysteines to take the reaction cycle 'full-circle'?
9. Figure legend 4: Please state how many technical/biological replicates were used in the graphs. Also, "bars" is an odd description. Was the mean plus/minus SD plotted?
10. Discussion, line 314: I-site inhibition in *Pseudomonas* WspR and GcbC uses two I-sites, creating pseudosymmetric dimers. Also, the BeF-activated state of PleD shows such a pseudosymmetric GGDEF:c-di-GMP complex. From the available structures, one may argue that such a dimer is the more prevalent form and not the outlier. Often, asymmetry within the dimer when considering the preceding domains is apparent as well. Unless I am misunderstanding the arguments the authors are making, these published results contradict the statements in the discussion. Please review this section and consider a comparison of DgcB with these structures, and alter the discussion accordingly.
11. All structures determined here show at least a dimer of FHA domains, suggesting a constitutive feature. Have the authors experimentally validated the oligomeric state of the full-length protein and isolated FHA domain (with and without phosphopeptide or crosslink)? This aspect is relevant also in light of the final model presented in Figure 5, which shows a monomeric enzyme as the

initial entity.

12. The authors discuss the presentation of a single peptide binding site in the full-length structure. Phosphopeptide binding experiments (maybe including an N-terminally truncated protein) could reveal binding stoichiometries. Such an experiment would reveal whether the activated complex opens up more than what the structures reveal so far. In contrast, sub-stoichiometric binding could have mechanistic consequences as well, as the authors have discussed.

We thank the reviewers for their time and skill in assessing our work, and provide a point-by-point response (marked with ####, in red) below. We have also highlighted these changes in the manuscript in red also.

Reviewers' comments:

Reviewer #1 (Remarks to the Author):

The manuscript entitled «Structural basis for activation of *Bdellovibrio* diguanylate cyclase that licenses prey entry» by Meek and co-workers, describes three interesting crystal structures of the cyclic-di-GMP synthase DgcB at high resolution. The analysis of the full-length and FHA-only structures enabled the determination of the FHA-only/phosphopeptide structure and ultimately the design of a constitutively active “disulfide-stapled” mutant comprising an engineered disulfide bridge. The work is technically sound and the manuscript is well written, albeit my personal impression is that the discussion section could be streamlined.

Although I find the work very interesting, particularly the resolutions are significantly better than the deposited structures of DgcB homologues, I am not excited about it. Particularly, I ask myself what do we really learn about the activation mechanism of this interesting family of enzymes (as the authors imply in the title)?

We respectfully disagree, agreeing with reviewer 2 that “the mechanistic underpinnings and in particular the regulation of these enzymes (is) less well defined.....provides interesting new data and models to the field, and more generally *Bdellovibrio*’s fascinating biology”. The key is not the

comparison to homologues regarding resolution BUT that the domain combination (FHA sensor) is unique and thus requires investigation of how activation occurs – we provide this.

Before our study, there was no clue as to the stimulus that activates DgcB (and thus predation); we have definitively identified this as recognition of a self phosphopeptide. We therefore present the first mechanistic analysis of a key control point in initiating predation.

Because we show the DgcB tail to be the site of phosphorylation, we are able (in the discussion) to implicate Bd3148 as the likely kinase that acts upstream. Excitingly, a new study using transposon mutagenesis to disrupt predation (Duncan *et al*, mBio, 2019, new ref #30) found the *bd3148* gene to be essential but gave no reason why. We have now worked this validating information into the discussion.

The full-length structure discloses little more than the structures of the isolated domains, because even the domain connectivity relies on distance constraints due to the lack of continuous electron density.

The domain connectivity is supported by three independent structures identifying identical FHA:FHA domain interactions, and secondly by the GGDEF:GGDEF domain interactions making biological sense (the I-sites coming together like that observed in pathway-unrelated *Pseudomonas* enzymes, Chen *et al* reference in manuscript). The full-length structure also discloses that in the resting state the tail is not ordered, which is the important observation that led us to uncover and validate the tail activation hypothesis.

The FHA-only structures confirm recognition of the phosphorylated N-terminus. For the *Bdellovibrio* enzyme, this is indeed new, but it is well-known that FHA domains recognize pThr residues as stated in reference #8 and shown in figure 2d.

The reviewer here misjudges the importance of discovering the precise nature of the activating peptide. Yes, one can predict that an FHA domain will ultimately bind a phosphopeptide, but it is essentially impossible to predict which peptide; here we identify and validate this for DgcB, which was (prior to our work) unknown. Existing fha:phosphopeptide complex structures are very similar (*e.g.* two of the best-studied GarA and OdhI are 70% identical), and so our study brings a valuable, diverse example to the literature. Furthermore, our covalent stapling of the stimulatory peptide brings something entirely new to the FHA field, and will be relevant for other studies in all domains of life.

Thus, the manuscript describes a solid piece of work, which agrees well with the current state of knowledge in the field, which is good, but after reading the title, I would expect at least some solid data on how peptide binding activates the catalytic domain.

Our data (obtaining both the unbound and bound states) are able to rule out conformational change within the FHA domain as the mode of activation – this is important because a significant proportion of signalling enzymes utilize this mechanism.

Forming the major part of our response, concerning both reviewers, we have carried out new experiments that now further inform on the likely mode of DgcB activation.

We have taken the active form of DgcB (cysteine-linked, I-site mutated), performed size exclusion chromatography to delineate this into monomers, dimers and higher-order oligomers, and assayed these independently for activity. This new data reveals that the dimeric fraction is highly active, the monomers/other oligomers less so. Supportive of this, our new non-reducing SDS-PAGE data reveals a single band corresponding to an intertwined dimer (tail activates neighbour FHA). We include these as a new figure that imposes strong constraints on the nature/mechanism of active DgcB.

The observation that the FHA:FHA interface seen in the full-length structure is conserved in the FHA-only structure (and perhaps in the FHA:phosphopeptide structure as well) is an interesting observation, particularly because the pocket is partially occupied by a bound ion. Can the authors say what kind of ion it is? Is this interaction biological relevant?

Due to the nature of crystallization conditions, this is likely to be Chloride. We do not believe this ionic interaction to be biologically relevant.

I would suggest to include a superposition of the FHA:FHA interface seen in all three structures, because I only learned from the discussion (page 16, line 377) that the contact is the same in all three structures.

We have followed this recommendation, and now include this as a new supplementary figure.

If this contact has any biological relevance, mutations of interface residues should alter the catalytic activity.

We made a mutation of the Asp residue that links FHA domains, and discovered that this didn't activate the protein – we are happy to include this as supplementary data if required. The reviewer suggests any mutation herein could alter catalytic activity, but the context of a multistage activation process could exclude this e.g. relieving inhibition (FHA contacts) is not the same as outright stimulation (our disulphide results), and hence might not influence activity strongly.

In figure 5c the authors assume that the biologically active DgcB species is a dimer and that the N-terminus can only bind intra- or inter-molecular. Is a daisy chaining event with subsequent oligomerization via FHA:N-terminus interactions unlikely?

As outlined above, we now show new data that reveals a continuum of oligomeric states, with the dimer being relatively the most active by far.

The same question arises for the hyperactive “disulfide-stapled” mutant. In principle, the presumed disulfide bridge could be intra- or inter-molecular. I would expect that both species have different activities and different retention times in size-exclusion chromatography. If there is only one dimeric species of this mutant a non-reducing SDS gel should reveal if the disulfide bridge is intra- or intermolecular.

We thank the reviewer for this suggestion and have performed this experiment, revealing the bridge to be intermolecular, forming part of the new composite figure 5.

Generally, I have the feeling that some sensible solution data would be very helpful for the proper analysis of these interesting crystal structures.

We agree, and the enzymatic assay of size-exclusion chromatography fractions now provides this solution data. As an addend, we also trialled SAXs experiments to determine solution scattering envelopes for different DgcB states, but the protein aggregated too much in these conditions for this approach to be informative.

Reviewer #2 (Remarks to the Author):

Bdellovibrio bacteriovorus is an organism that consumes other bacterial cells in a complex lifecycle that involves the bacterial second messenger c-di-GMP. *Bdellovibrio* encodes several enzymes that make and break c-di-GMP in addition to an abundance of potential c-di-GMP receptors. While genetic and cellular studies have assigned distinct functions of many of the enzymes involved in c-di-GMP signaling, the mechanistic underpinnings and in particular the regulation of these enzymes remained less well defined.

We thank reviewer 2 for noting this deficit prior to our study.

The manuscript by Meek et al. describes a thorough structure-function analysis of one such enzyme, a unique diguanylate cyclase (DGC) of *Bdellovibrio*. The protein, DgcB, is comprised of a sensory FHA domain flanked by an N-terminal tail and a C-terminal GGDEF domain. Based on three novel, insightful crystal structures (full-lengths, isolated FHA domain, and isolated FHA domain bound to phosphopeptide), the authors derive the central hypothesis that the N-terminal peptide of DgcB itself becomes the activating signal of the FHA domain when phosphorylated. They go on to test this model by structure-informed engineering of a hyper-active enzyme. The study is rigorous and provides interesting new data and models to the c-di-GMP field, and more generally, to the regulation of *Bdellovibrio*'s fascinating biology.

Again, we thank the reviewer for appreciation of our work.

There are only a few minor points the authors should consider when revising the manuscript. Many of the points can be addressed by editing the text. Some key experiments would add value to the current study; they should be feasible, making use of available reagents and the expertise of the investigators:

1. Introduction: The first paragraph could use some key references to primary literature. While the review cited in the first sentence covers most of what is described, explicit mentioning of key publications would be a welcome addition.

We have followed the reviewer's advice and added references to primary studies that outline several statements in the opening paragraph (Ryjenkov *et al* experimental proof of PilZ function; Sudarsan *et al* discovery of riboswitches; Lee *et al* discovery of degenerate enzyme as receptor).

2. Introduction, line 62: Is there is a reason why the number of PilZ proteins cannot be determined precisely?

Yes, some of these are degenerate and so can't be identified as easily as consensus family members. Further to this, *Bdellovibrio* gene sequences can be quite divergent from "regular" bacteria.

3. Throughout the manuscript but particularly in the introduction: The excessive use of texts in brackets has a somewhat negative impact on readability.

We apologise and have attempted to edit regions of the manuscript to correct this – p3, p4, twice, p5, p11, p16, p18, p20

4. Figures (all): Please increase the label font size in the figure panels. Many are hard to read in a printed version of the figures.

These should suffice in final versions – editorial staff can advise if necessary.

5. Line 224: The authors describe the FHA:tail interaction as occurring "in-cis". In the most stringent meaning, this would imply that the peptide tail comes from the same polypeptide chain as the FHA domain. Based on the previous structural description in the manuscript, this argument cannot be made conclusively. To stay on the safe side, I would rephrase this sentence so that it includes the possibility of the tail reaching over from an adjacent DgcB molecule (in a dimeric/multimeric assembly).

We state that both possibilities may occur; we have interpreted the new SDS-PAGE data as identifying an inter-linked dimer and have noted this increased likelihood in the text.

6. Line 231: The authors only describe phosphopeptide binding to chain B of the complex structure. How does chain A compare? If chain A lacks clear indications of peptide binding, please comment on the possible reasons for non-equivalent protomers and whether you think that such a difference could have mechanistic implications.

The solubility of peptide is sparingly low, and could have been further reduced in-crystallo. The inability to monitor stoichiometry during crystallogenesis means it is best not to read too much into the final 1:2 ratio. We do not believe the non-equivalence to be relevant, and have added a statement to the main text (p19).

7. Figure 3: Have the authors attempted to determine the affinity and stoichiometry of phosphorylated vs unphosphorylated tail peptide binding to the FHA domain? This is only a minor point since the interaction could occur in cis or within a complex, and as a result the affinities of free peptide may only approximate the interaction strength within the full-length protein.

We suspect that the reviewer is correct on the difficulty of local clustering affecting this type of measurement – we tried this via ITC but were unable to measure a definitive affinity.

8. Figure 4: In an elegant approach,

We thank the reviewer for appreciating the usage of the disulphide experiments.

the authors engineer a disulfide bond between FHA domain and the N-terminal tail that vastly increases DGC activity of the enzyme. Have the authors tried to add phosphorylated peptide in trans to full-length of N-terminally truncated DgcB? Such an experiment appears feasible and would add to the mechanistic studies as it could present activation by the native ligand.

We did try adding exogenous peptide in our activity assays and it was unable to activate, indicating that tail-sensor-enzyme context in the full-length protein is an important factor in the mechanism of regulation. Again, this fits our structural data that indicate a lack of observable conformational change in FHA domain upon binding. We are happy to include this exogenous peptide activity data in supplementary if required.

Similarly, the TCEP reduction experiment is neat as it shows reversibility of the disulfide bond effect. I wonder if the authors have attempted to re-oxidize the cysteines to take the reaction cycle ‘full-circle’?

We thank the reviewer for appreciating the reversibility; we did not take this full circle as (i) the timescale would be too long in relation to protein stability and (ii) it wouldn't change the interpretation of results or inferred mechanism.

9. Figure legend 4: Please state how many technical/biological replicates were used in the graphs. Also, “bars” is an odd description. Was the mean plus/minus SD plotted?

We used 3 technical repeats and have now included this information in the legend; SD was used for hplc, standard error for CD.

10. Discussion, line 314: I-site inhibition in Pseudomonas WspR and GcbC uses two I-sites, creating pseudosymmetric dimers. Also, the BeF-activated state of PleD shows such a pseudosymmetric GGDEF:c-di-GMP complex. From the available structures, one may argue that such a dimer is the more prevalent form and not the outlier. Often, asymmetry within the dimer when considering the preceding domains is apparent as well. Unless I am misunderstanding the arguments the authors are making, these published results contradict the statements in the discussion. Please review this section and consider a comparison of DgcB with these structures, and alter the discussion accordingly.

We apologise for any confusion here – the DgcB I-site interaction is essentially symmetric (in terms of GGDEF arrangement) but pseudosymmetric when the FHA domains below are considered. The PleD I-site arrangement is different, using two differing pockets, one of which is a conventional I-site. We hope that the use of “both” (line 10 of paragraph) vs single (line 12) communicates this clearly.

11. All structures determined here show at least a dimer of FHA domains, suggesting a constitutive feature. Have the authors experimentally validated the oligomeric state of the full-length protein and isolated FHA domain (with and without phosphopeptide or crosslink)? This aspect is relevant also in light of the final model presented in Figure 5, which shows a monomeric enzyme as the initial entity.

Our new data, show that an isolated FHA domain (missing linker and GGDEF) is a monomer in solution. The I-site mutated protein is also a monomer. Hence, dimerization may result from local clustering, either from high local concentration in a crystal, I-site occupation by cyclic-di-GMP bringing FHA together, or via DgcB activation upon phosphopeptide binding.

12. The authors discuss the presentation of a single peptide binding site in the full-length structure. Phosphopeptide binding experiments (maybe including an N-terminally truncated protein) could reveal binding stoichiometries.

Such an experiment would reveal whether the activated complex opens up more than what the structures reveal so far. In contrast, sub-stoichiometric binding could have mechanistic consequences as well, as the authors have discussed.

####As outlined above, attempts to do this via ITC were unsuccessful.

REVIEWERS' COMMENTS:

Reviewer #2 (Remarks to the Author):

The authors addressed my questions satisfactorily.